# Farnesyltransferase inhibition overcomes oncogene-addicted non-small cell lung cancer adaptive resistance to targeted therapies

Sarah Figarol [1,7], Célia Delahaye [1,7], Rémi Gence[1], Aurélia Doussine[1], Juan Pablo Cerapio [1], Mathylda Brachais[1], Claudine Tardy[1], Nicolas Béry [1], Raghda Asslan[1], Jacques Colinge [2], Jean-Philippe Villemin[2], Antonio Maraver [2], Irene Ferrer [3], Luis Paz-Ares[3], Linda Kessler[4], Francis Burrows[4], Isabelle Lajoie-Mazenc[1], Vincent Dongay[1,5], Clara Morin[1,5], Amélie Florent[1], Sandra Pagano [1], Estelle Taranchon-Clermont[1,6], Anne Casanova[6], Anne Pradines[1,6], Julien Mazieres [1,5], Gilles Favre [1,6,7] ✉ & Olivier Calvayrac [1,7] ✉

Drug-tolerance has emerged as one of the major non-genetic adaptive processes driving resistance to targeted therapy (TT) in non-small cell lung cancer (NSCLC). However, the kinetics and sequence of molecular events governing this adaptive response remain poorly understood. Here, we combine real-time monitoring of the cell-cycle dynamics and single-cell RNA sequencing in a broad panel of oncogenic addiction such as EGFR-, ALK-, BRAF- and KRAS-mutant NSCLC, treated with their corresponding TT. We identify a common path of drug adaptation, which invariably involves alveolar type 1 (AT1) differentiation and Rho-associated protein kinase (ROCK)-mediated cytoskeletal remodeling. We also isolate and characterize a rare population of early escapers, which represent the earliest resistance-initiating cells that emerge in the first hours of treatment from the AT1-like population. A phenotypic drug screen identify farnesyltransferase inhibitors (FTI) such as tipifarnib as the most effective drugs in preventing relapse to TT in vitro and in vivo in several models of oncogenic addiction, which is confirmed by genetic depletion of the farnesyltransferase. These findings pave the way for the development of treatments combining TT and FTI to effectively prevent tumor relapse in oncogene-addicted NSCLC patients.

Targeted therapies (TT) have revolutionized the therapeutic management of patients harboring non-small cell lung cancer (NSCLC) with genetic alterations on oncogenic drivers such as epidermal growth factor receptor (EGFR)[1,2], anaplastic lymphoma kinase (ALK)[3,4], ROS proto-oncogene 1 (ROS1)[5,6], human epidermal growth factor receptor-2 (HER2)[7], v-raf murine sarcoma viral oncogene homolog B1 (BRAF)[8], MET proto-oncogene (MET)[9], or more recently Kirsten rat sarcoma viral oncogene homolog (KRAS)[10,11]. However, despite these advances,

TTs are rarely curative and nearly all patients develop resistance within a relatively short period. While extensive research has been conducted to elucidate the mechanisms underlying resistance, the multitude of genetic and non-genetic molecular events identified in relapsing tumors strongly limits the therapeutic alternatives for patients at the time of recurrence[12–15]. Despite the successful targeting of some identified resistance mechanisms (e.g., third-generation EGFR tyrosine kinase inhibitor (TKI) osimertinib which efficiently targets the EGFR[T790M] resistance mutation under first-generation EGFR-TKI erlotinib or gefitinib[16]), the emergence of additional resistance mechanisms systematically undermines the effectiveness of this strategy[17]. This prompted the scientific community to delve into the origins of these resistances, initially addressing the pivotal question of whether resistant clones preexisted within tumors prior to treatment or if they emerged de novo in response to the drugs[18]. Although both mechanisms might coexist within tumors[19], the identification of a so-called drug-tolerant persister (DTP)[20] or drug-tolerant cell (DTC) population, along with the large number of studies that arose from this discovery[21–28], strongly suggest that drug adaptation through phenotypic plasticity of tumor cells may play a major role in the development of both genetic and non-genetic resistance mechanisms.

DTCs are defined as a phenotypically heterogeneous population of slow-to-non proliferating cells that emerge through stochastic selection after high drug exposure, with no evidence of known resistance mechanisms (recently reviewed[29]). DTCs have been most extensively studied using the EGFR-mutant PC9 lung adenocarcinoma cell line treated with EGFR-TKI[19–22,24,27,28], but have also been described in a broader spectrum of cancers including metastatic melanoma[30], glioblastoma[31], colorectal[32], breast[33], or pancreatic cancer[34]. DTCs have been shown to display epigenetic[20,27] and metabolic[35] alterations as well as transcriptional[28] and translational[36] reprogramming. In light of these observations, various strategies have been proposed to eradicate DTCs, with a specific focus on those arising from EGFR-mutant NSCLC treated with EGFR-TKIs. These approaches include combinations of EGFR-TKIs with epigenetic modulators (e.g., histone deacetylase (HDAC)[20,27] or H3K4 demethylase KDM5[37] inhibitors), ferroptosis-inducing agents (e.g., glutathione peroxidase GPX4 inhibitor RSL3[24]), cyclin-dependent kinase (CDK) inhibitors (e.g., CDK7/12 inhibitor THZ1[38]), Aurora kinase A (AURKA)[39] and Aurora kinase B (AURKB)[40] inhibitors, or AXL inhibitors[25,41], amongst others. Despite promising results in preclinical models, none of these combinations are currently approved for clinical use, which may be explained, at least in part, by an incomplete understanding of the molecular mechanisms underlying the drug-tolerant state. As an example, recent findings indicate that while RSL3-induced ferroptosis effectively reduces the overall amount of DTCs, it paradoxically amplifies the proportion of cycling persisters, a cell population thought to contribute to the emergence of resistant clones[35], thus highlighting the importance of considering the dynamic nature of the drug-tolerant state to develop effective therapeutic strategies.

Here, we report an in-depth phenotypic and molecular characterization of the early events leading to drug resistance using EGFR-mutant NSCLC as a reference model, and extending our findings to other oncogenic settings such as ALK[EML4], BRAF[V600E], or KRAS[G12C] NSCLC. We establish an innovative approach by combining real-time monitoring of the cell cycle dynamics and single-cell RNA sequencing, leading to the identification of hallmarks and vulnerabilities of DTC that can be pharmacologically targeted using farnesyltransferase inhibitors in several preclinical models of oncogene-addicted NSCLC, providing further rationale for advancing these combination therapies into clinical development.

## Results

### Drug tolerance is a dynamic rather than a dormant state

Although an increasing number of studies have focused on the characterization of DTC and fully resistant proliferative cells

(RPC)[19–22,27,35,42], the kinetics of evolution through the different states is largely unknown. We used the FUCCI (fluorescence ubiquitination cell cycle indicator) system[43] to perform real-time monitoring of the cell cycle dynamics in a panel of EGFR-mutant NSCLC cell lines treated with 1 μM EGFR-TKI (erlotinib or osimertinib). Importantly, cell lines were previously subcloned to minimize the presence of potential pre-existing resistant cells[19,44] (Supplementary Fig. 1A–B). For all cell lines, we observed a common pattern of G1 accumulation within the first 48 h of treatment (Fig. 1a, b, Supplementary Fig. 2A, Supplementary Movies 1–4), which was invariably associated with p27[Kip1] overexpression and Retinoblastoma (Rb) protein dephosphorylation (Fig. 1c, Supplementary Fig. 2B). This pattern was also observed in other NSCLC models of oncogenic addiction such as KRAS[G12C], ALK[EML4], or BRAF[V600E], treated with their respective targeted therapies (i.e., sotorasib, lorlatinib and dabrafenib) (Supplementary Fig. 3A–C). While most cells remained in G1 and progressively died resulting in a bulk population decrease, a subset of cells, referred to as "early escapers", rapidly progressed through S/G2 (Fig. 1a–b, Supplementary Fig. 2A, Supplementary Fig. 3A–C, Supplementary Movies 1–4), an observation consistent with a rare, drug-tolerant cycling subpopulation as recently described[35].

To identify the molecular mechanisms underlying evolution from G1-arrest to early escape, we performed scRNAseq on ~3000 G1 (red) and S/G2 (green) cells sorted from both untreated and osimertinib-treated HCC4006 cells, which enabled the enrichment of the rare population of early escapers (Supplementary Fig. 4A, B). The seurat analysis identified different clusters, which were mostly related to the position of cells within the cell cycle (Fig. 1d, Supplementary Fig. 4C). In line with previous reports[19,22], myogenesis and epithelial-to-mesenchymal (EMT) signatures were strongly upregulated in both G1- and S/G2-treated cells (Supplementary Fig. 4D), while cell cycle-related gene signatures such as E2F_targets or G2M_checkpoint were profoundly downregulated in G1 and fully restored in early escapers (Fig. 1e, Supp Fig. 4C, D). We observed a deep lineage reprogramming during the adaptive response, which involved a robust repression of mucous/serous-related genes (e.g., PIGR, BPIFB1/2, SCGB3A1, or MUC5B) and a progressive acquisition of mesenchymal features in early escapers, consistent with an EMT process (Fig. 1e, f, Supplementary Fig. 4E–G). Interestingly, one of two osimertinib-treated G1 clusters (osi-G1[#4]) was highly enriched in alveolar type 1 (AT1)-gene signature[45] (e.g., AQP4, CYP4B1, CLIC5, AGER or TNNC1), while the second G1 cluster (osi-G1[#5]) was enriched in mesenchymal-related genes (e.g., SERPINE1, ADAM12, SPOCK1, or MATN2) (Fig. 1e, f, Supplementary Fig. 4E–G). AT1-specific markers were upregulated early during treatment and were restricted to the non-cycling drug-tolerant population, whereas mesenchymal features were specifically increased in resistant proliferative cells, suggesting that tumor cells may have undergone an alveolar-like differentiation process prior to the acquisition of a mesenchymal phenotype (Fig. 1g). The AT1-specific marker AGER was also transiently upregulated during drug-tolerance in other models of oncogenic addiction such as KRAS[G12C], BRAF[V600E] or ALK[EML4] treated with their corresponding TT, and was lost in resistant proliferative cells (Supplementary Fig. 4H). Pseudotime analysis revealed a tight connection between AT1 (osi-G1[#4]) and mesenchymal (osi-G1[#5]) clusters, which suggests that mesenchymal-like cells could have evolved from the alveolar-like population (Fig. 1h, Supplementary Fig. 5A–C). Early escapers sorted after only 5 days of osimertinib treatment immediately reproliferated in the presence of the drug, confirming that these cells had already acquired a resistance mechanism, whereas G1-sorted cells displayed a latency of two-to-three weeks before developing resistant proliferative cells, showing that this population may constitute a reservoir from which resistant cells could emerge (Fig. 1i).

We next aimed to determine the molecular pathways involved in the acquisition of the different phenotypes. We correlated the mean

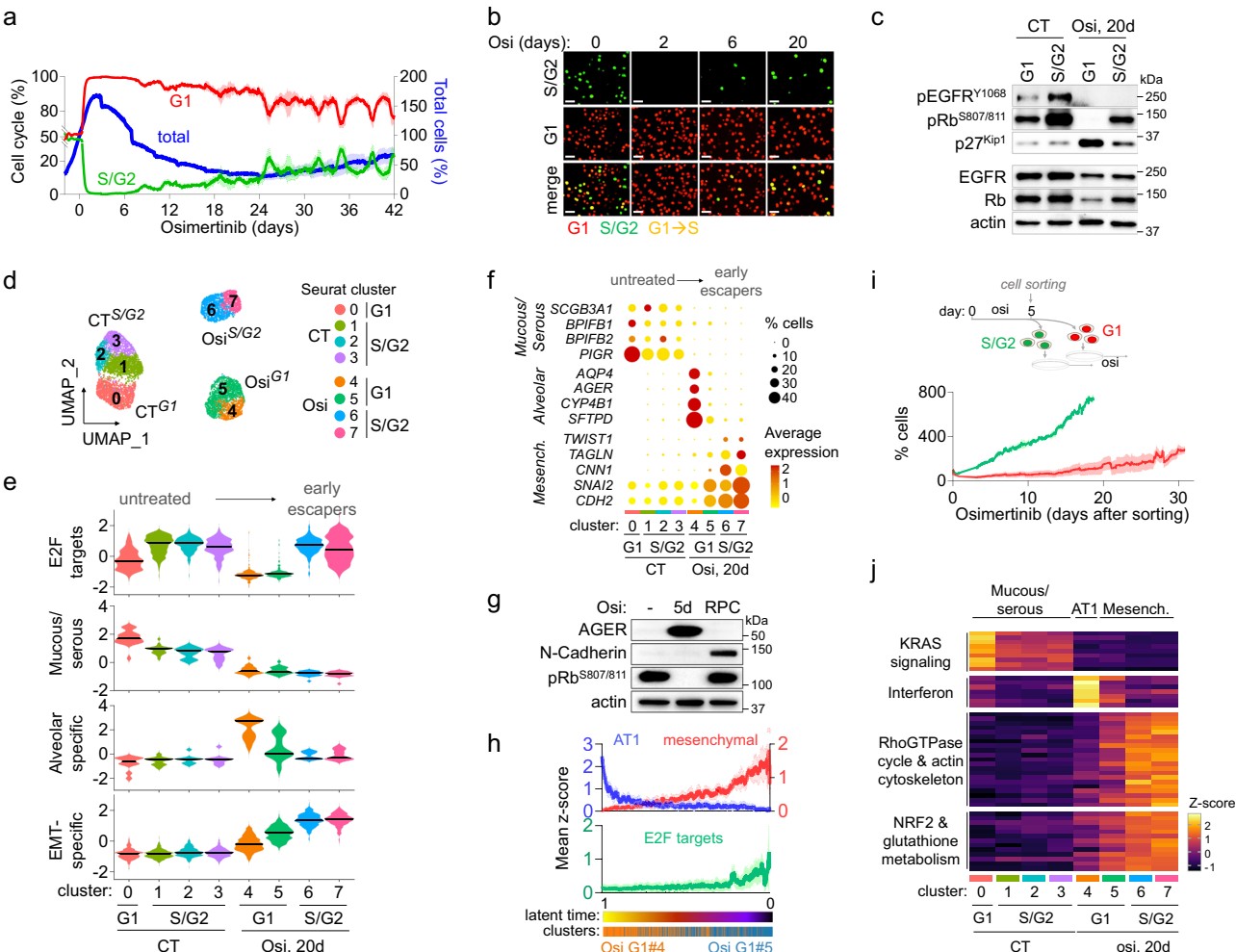

**Fig. 1 | Drug tolerance is a dynamic state which involves a multistep phenotypic reprogramming. a** Percentage of total (blue), S/G2 (green) or G1 (red) populations of HCC4006 subclonal cells during osimertinib treatment (1 μM). Data are mean ± SEM. Representative data from $n = 4$ independent biological experiments. **b** Representative fluorescence images of S/G2 (green) or G1 (red) HCC4006 subclonal cells during osimertinib treatment (1 μM) from $n = 4$ independent biological experiments. Scale bar: 50 μm. **c** Western Blot analysis of EGFR, retinoblastoma (Rb) and p27$^{Kip1}$ signaling pathway in untreated (CT) and osimertinib-treated S/G2 or G1 HCC4006 subclonal cells. Representative blots from $n = 2$ independent biological experiments. **d** UMAP plot of the different clusters from Seurat analysis of untreated (CT) and osimertinib-treated HCC4006 subclonal cells obtained after scRNAseq. The relative percentage of each population was G1 = 35.5% and S/G2 = 30.3% for untreated cells and G1 = 70.4% and S/G2 = 0.6% after 20 days of osimertinib treatment 1 μM. Between 2000 and 3000 cells were recovered for each condition for library preparation. **e** Violin plots representing the distribution of indicated signature scores in the different clusters. **f** Dot plot showing the expression level and percentage of expressing cells of genes specific for the mucous/serous, alveolar and mesenchymal phenotypes in the different clusters. **g**, Western Blot analysis of proteins related to the alveolar (AGER) and mesenchymal (N-Cadherin) phenotypes in untreated, osimertinib-treated for 5 days and osimertinib-resistant proliferative HCC4006 subclones (RPC: Resistant Proliferative cells). Representative blots from $n = 4$ independent biological experiments. **h** Distribution of the mean normalized (z-score) expression of alveolar (blue), mesenchymal (red) and E2F targets (green) signatures, based on latent time. Osi-G1 cluster 4 is shown in orange and cluster 5 in blue. Data are mean ± SEM. **i** Proliferation of G1 (red) and S/G2 (green) HCC4006 subclonal cells sorted by FACS after 5 days of osimertinib treatment (1 μM) and re-plated in the presence of the drug. Data are mean ± SEM. Representative data from $n = 3$ independent biological experiments. **j** Mean normalized (z-score) expression per cluster of genes involved in KRAS signaling, interferon, Rho GTPase cycle, actin cytoskeleton, NRF2 and glutathione metabolism pathways. Source data are provided as a Source data file.

expression of each gene with the expression of the most representative gene for each phenotype (e.g., *AGER* for the AT1 phenotype, *SERPINE1* for the mesenchymal phenotype, and *BPIFB1* for the mucous/serous phenotype; Pearson correlation coefficient > 0.9) (Supplementary Fig. 6A, D, G, Supplementary Data 1). We determined that the mucous/serous phenotype was mostly associated with KRAS signaling-related genes (Fig. 1j, Supplementary Fig. 6A–C, Supplementary Data 1), the alveolar phenotype was highly enriched in interferon-related genes (Fig. 1j, Supplementary Fig. 6D–F, Supplementary Data 1), and the mesenchymal-associated phenotype was strongly associated with EMT, RHO_GTPASE_CYCLE and ACTIN_CYTOSKELETON signatures, as well as some NRF2 (*NFE2L2*) and glutathione metabolism-related genes (Fig. 1j, Supplementary Fig. 6G–K, Supplementary Data 1).

Altogether, our data show that targeted therapies invariably induce a rapid cell cycle arrest characterized by the activation of the p27$^{Kip1}$/pRb pathway, followed by the emergence of a rare population of proliferative early escaper cells. The drug-tolerant state displayed two distinct although tightly linked populations composed by an alveolar-like subpopulation enriched in interferon-related genes that were restricted to the non-proliferative state, and a mesenchymal subpopulation that was mostly associated with early escapers and was characterized by an increased expression of genes associated with Rho-GTPase activity and actin cytoskeleton remodeling. The sequential emergence of the two phenotypes, first alveolar and then mesenchymal, combined with pseudotime analysis, suggest that mesenchymal cells may have evolved from the alveolar-like population. However, we

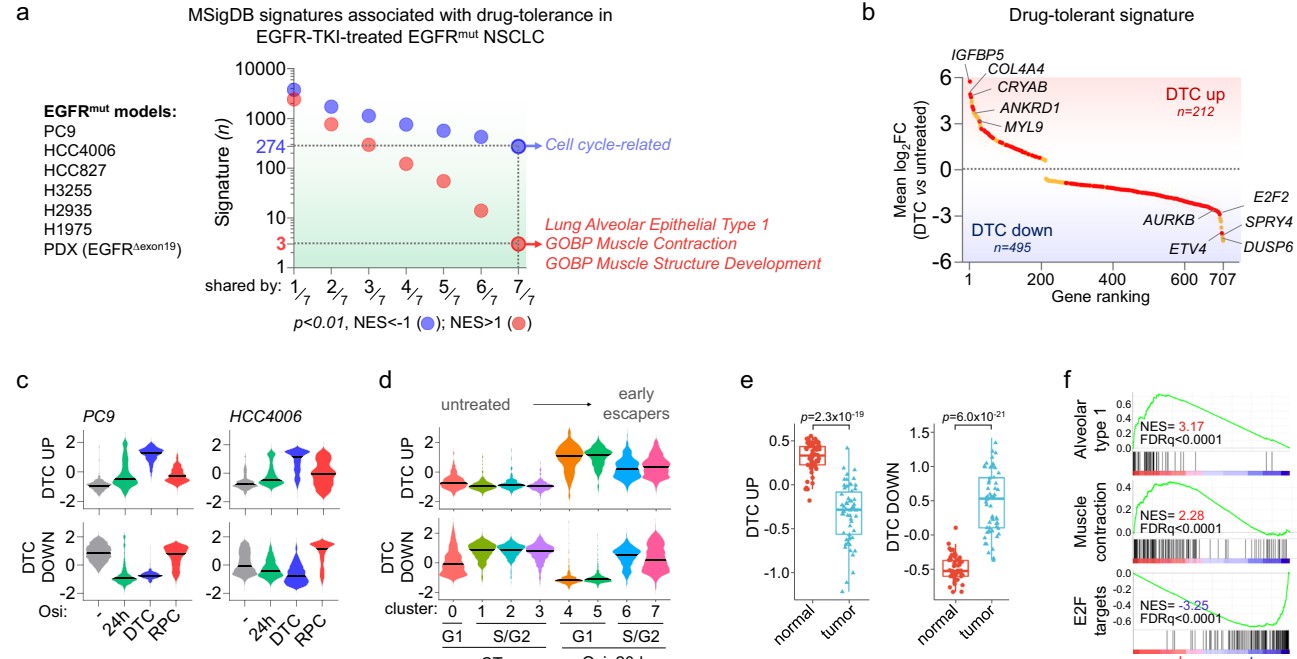

**Fig. 2 | Drug-tolerant cells transcriptomic signature reveals similarities with normal alveolar cells. a** GSEA analysis showing downregulated (blue; nominal *p*-value < 0.01, NES < −1) and upregulated (red; nominal *p*-value < 0.01, NES > 1) gene signatures from the Human Molecular Signatures Database (MSigDB), organized from the least frequently shared (1/7) to the most frequently shared (7/7) signatures across the different models of EGFR-TKI-induced drug-tolerant cells. Nominal *p*-value and NES were calculated using GSEA software. **b** Drug-tolerant-associated genes ranked based on the mean log2 fold change expression (DTC *vs* untreated, *p* < 0.01) in 7/7 (red) or 6/7 (orange) models described in a. Data are mean ± SEM. *p*-value was obtained by DESeq2 analysis. **c** and **d** Distribution of mean expression levels (z-scores) of genes from the DTC_UP and DTC_DOWN signatures in untreated cells or at indicated stages of treatment in PC9 and HCC4006 subclones (**c**) and in the different clusters identified by scRNAseq (**d**). **e** Box plots of signature scores of DTC_UP and DTC_DOWN in lung adenocarcinomas (tumor, *n* = 58) compared to adjacent normal lung tissue (normal, *n* = 58) from TCGA-LUAD database. The box plots display 25th (lower bound), 50th (center, median), and 75th (upper bound) percentiles, with whiskers generated with the Tukey method; all points are shown. *p*-value was calculated using a two-tailed paired *t*-test. **f** GSEA analysis of alveolar type 1, muscle-contraction and E2F targets signatures in normal lungs compared to lung adenocarcinomas using the TCGA-LUAD database. DTC: drug-tolerant cell; RPC: resistant proliferative cell. Source data are provided as a Source data file.

cannot exclude that tumor cells could have undergone distinct paths of drug adaptation toward both phenotypes during the adaptive response.

## Drug-tolerant cells transcriptomic signature reveals similarities with normal alveolar cells

Since these findings could be specific to the HCC4006 model, we performed bulk RNAseq experiments in drug-tolerant cells generated in three other EGFR-mutated cell lines treated with either erlotinib or osimertinib (PC9, HCC827, H3255), and we compared their transcriptomic profiles to other recently published transcriptomes of osimertinib-induced drug-tolerance, which included two other cell lines (HCC2935 and H1975)[46] and one EGFR-mutant NSCLC PDX model[23] (Fig. 2a, Supplementary Table 1). Among 27,864 Human Molecular Signatures Database (MSigDB) gene sets tested, only 3 signatures were commonly upregulated in all the models (NOM *p*-value < 0.05), namely TRAVAGLINI_LUNG_ALVEOLAR_EPITHELIAL_TYPE_1_CELL, and two muscle-related gene signatures (GOBP_MUSCLE_CONTRACTION and GOBP_MUSCLE_STRUCTURE_DEVELOPMENT), with some overlapping genes between those signatures such as *MYL9, CRYAB, TAGLN, ATF3,* or *TNNC1* (Fig. 2a, Supplementary Fig. 7A–C, Supplementary Data 2). On the other hand, 274 signatures were commonly downregulated and were mostly associated with the cell cycle (Fig. 2a, Supplementary Fig. 7A, Supplementary Data 2). Using these transcriptomic data, we built a new signature of drug tolerance composed of 212 genes commonly upregulated (i.e., *p* < 0.01, log2FC > 0.5 in at least 6 out of 7 models; DTC_UP) and 495 genes commonly downregulated (i.e., *p* < 0.01, log2FC < −0.5 in at least 6 out of 7 models; DTC_DOWN) (Fig. 2b,

Supplementary Data 3). Consistent with previous analysis, DTC_UP signature was enriched in genes related to alveolar type 1, EMT, muscle contraction, extracellular matrix (ECM), insulin-like growth factors (*IGFBPs*) and RHO-GTPase_cycle (Supplementary Fig. 8A). This signature was highly specific of the drug-tolerant state and was almost completely reversed in fully resistant proliferative cells (RPC) (Fig. 2c). Consistently, DTC_UP signature was upregulated in both osi-G1[#4] and osi-G1[#5] clusters and was partially reversed in early escapers (Fig. 2d). Interestingly, both DTC_UP and DTC_DOWN signatures were strongly associated with EGFR-TKI-induced drug-tolerance in vivo[23] but also in other NSCLC models of oncogenic addiction such as KRAS[G12C][47], ALK[EML4][48], or BRAF[V600E][49] treated with their respective targeted therapies, suggesting a common path of drug adaptation (Supplementary Fig. 1B). DTC_UP signature was also associated with healthy lungs compared to lung adenocarcinomas, with 136/212 (64.2%) genes significantly upregulated in both DTC and normal lungs (Fig. 2e, Supplementary Fig. 9A), while 395 of the 495 (79.8%) genes of the DTC_DOWN signature where significantly downregulated in normal tissue *versus* lung tumors (Fig. 2e, Supplementary Fig. 9B), strongly reinforcing the evidence that DTC shared phenotypic resemblance with normal epithelial lung cells. Nevertheless, some genes were specifically upregulated in DTC (*vs* untreated) but not in healthy lungs (*vs* tumors) such as *IGFBP5, SPARC, RHOBTB3, SOX4, L1CAM* or *TRIO*, among others (Supplementary Fig. 9A). Conversely, some genes were specifically downregulated in response to EGFR-TKI but not in healthy lungs (*vs* tumors), such as *DUSP6, ETV5, SPRY4, PPARG, SCD* of *LDLR*, among others (Supplementary Fig. 9B), highlighting differences between drug-tolerant cells and healthy cells. As in DTC, genes

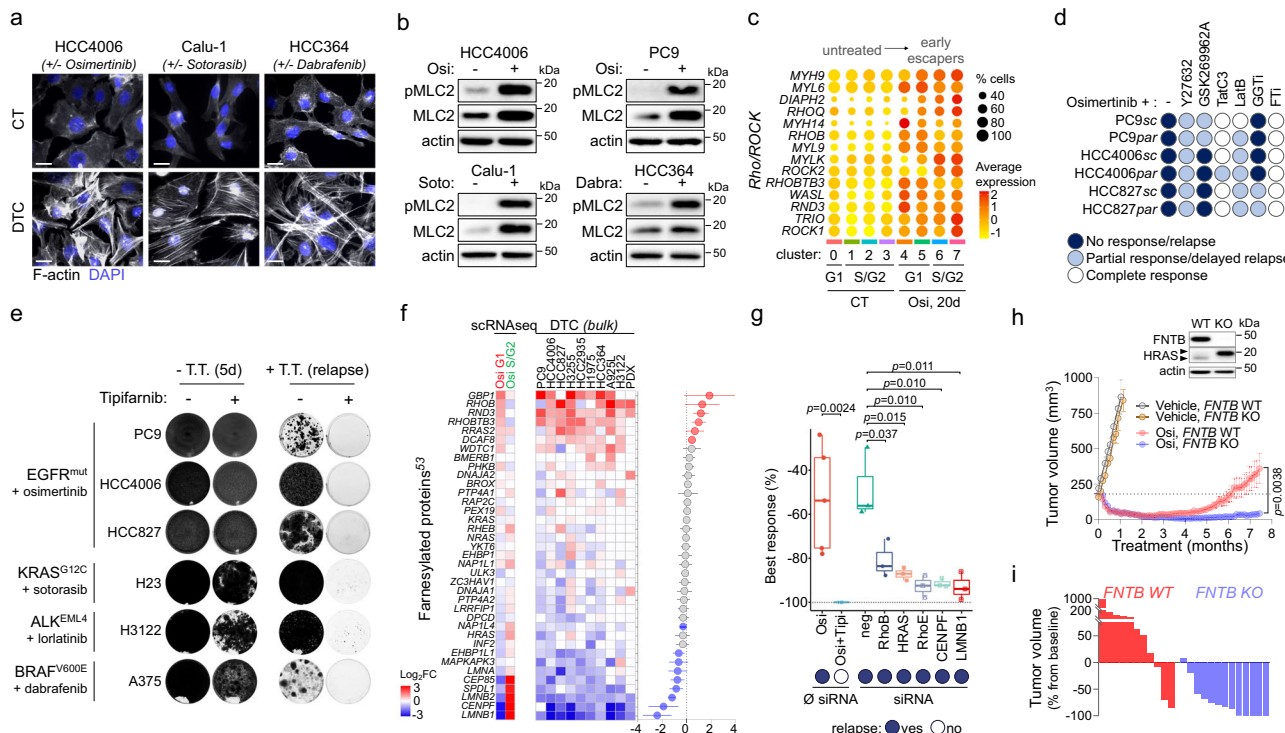

**Fig. 3 | Farnesyltransferase inhibition prevents the emergence of resistance to targeted therapies in vitro. a** Filamentous actin (F-actin) staining of indicated NSCLC cell line models in control or drug-tolerant cells (DTC) treated with their corresponding targeted therapy at 1 μM. Scale bar: 20 μm. Representative of *n* = 3 independent biological experiments. **b** Western blot analysis of total and phosphorylated MLC2 in the indicated NSCLC cell line models in control cells or DTC treated with their corresponding targeted therapy at 1 μM. Representative blots from *n* = 3 independent biological experiments. **c** Bubble plot showing the expression and distribution of genes involved in the Rho/ROCK pathway in the different clusters. **d** Cellular response of PC9, HCC4006, HCC827 parental and subclonal cells treated with 1 μM osimertinib alone or in combination with ROCK1/2 inhibitors (Y276324, 10 μM; GSK269962A4, 5 μM), RHOA/B/C inhibitor (TatC3, 5 μg/ml), actin polymerization inhibitor (LatB, Latruncunlin B, 0.3 μM), geranylgeranyltransferase inhibitor (GGTi, 1 μM) or farnesyltransferase inhibitor (FTi; 1 μM). Data are representative of *n* = 3 independent biological experiments. **e** Crystal violet staining of indicated cellular models untreated or treated until relapse with corresponding targeted therapy (T.T.) alone (1 μM) or in combination with Tipifarnib (1 μM). Representative images from *n* = 3 independent biological experiments. **f** Differential expression of genes (*p* < 0.01) coding for farnesylated proteins[53] in osi-G1 *vs* CT-G1 (G1) and osi-S/G2 *vs* osi-G1 (S/G2) in HCC4006 subclonal cells (left) and

in indicated NSCLC models of drug-tolerance (Right); *p*-value for each individual model was obtained by DESeq2 analysis. Right: mean (Log2FC [DTC *vs* untreated], all models) ±SEM; red dots: upregulated with *p* < 0.01, blue dots: downregulated with *p* < 0.01, gray: non-significant; *p*-value for mean Log2FC was calculated using two-tailed unpaired *t*-test. **g** Box plots showing the percentage of best response after treatment with osimertinib alone (1 μM) or in combination with Tipifarnib (1 μM), or after transfection with indicated siRNA (10 nM). The box plots display 25th (lower bound), 50th (center, median), and 75th (upper bound) percentiles, with whiskers generated with the Tukey method; Data represent *n* = 5 (osi, osi+tipi) or *n* = 3 (siRNA) independent biological experiments. *p*-value was calculated using a two-tailed unpaired *t*-test. **h** Top: Western blot analysis of FNTB protein expression and HRAS prenylation status in the FNTB WT and KO cells. The upper and lower arrows show unfarnesylated and farnesylated proteins respectively. Bottom: Mean tumor volume of PC9 FNTB-WT and FNTB-KO xenografts. Mice were treated 5 days/week with vehicle or Osimertinib (Osi, 5 mg/kg, q.d). Vehicle WT, *n* = 3; Vehicle KO, *n* = 3; osi WT, *n* = 12; osi KO, *n* = 14. Data are mean ± SEM; the *p*-value was calculated using two-tailed unpaired *t*-test. **i** Percentage of tumor volume from baseline after 7 months of osimertinib treatment in mice bearing FNTB-WT and FNTB-KO PC9 xenografts. Source data are provided as a Source data file.

overexpressed in healthy tissues compared with lung adenocarcinomas correlated with AT1 and muscle contraction-related signatures (Fig. 2f, Supplementary Fig. 10A, B), and displayed enrichment for cytoskeletal-related genes (Supplementary Fig. 10A–C, Supplementary Table 2).

Overall, we defined a transcriptomic signature of DTC, which revealed similarities with normal alveolar cells mainly characterized by increased contractility and cytoskeletal remodeling.

### Farnesyltransferase inhibition prevents the emergence of resistance to targeted therapies in NSCLC

We next aimed to assess the origin and the biological consequences of increased contractile gene signatures. We observed a highly reorganized actin cytoskeleton in all the EGFR-TKI-induced DTC evidenced by the presence of actin stress fibers and/or lamellipodia, which became visible after 3 to 5 days of treatment (Fig. 3a, Supplementary Fig. 11A). Stress fibers were also observed in other oncogenic

settings such as KRAS[G12C] or BRAF[V600]-mutant NSCLC cells treated respectively with sotorasib and dabrafenib (Fig. 3a), suggesting that actin cytoskeleton remodeling was a common cellular response to targeted therapy treatment. Since the presence of F-actin often correlates with Rho/ROCK pathway activation[50,51], and given the association of DTC with the RHO_GTPASE_CYCLE signature (Fig. 1j, Supplementary Fig. 8A), we hypothesized that this pathway could preferentially mediate the phenotypic plasticity observed during the adaptive response to targeted therapy. Consistent with this hypothesis, we observed an increase in both total and phosphorylated myosin light chain 2 (pMLC2) (Fig. 3b), as well as increased expression of several Rho/ROCK-related genes in both G1 and S/G2 osimertinib-treated cells (Fig. 3c, Supplementary Fig. 11B, C). It is noteworthy that overexpression of *MYL9* (MLC2-coding gene) was observed in both the alveolar-like and mesenchymal subpopulations (Fig. 3c, Supplementary Fig. 11D), reinforcing the phenotypic link between both subpopulations.

Based on these observations, we aimed to determine whether pharmacological inhibition of this pathway could prevent the emergence of resistance to EGFR-TKI. We tested a panel of inhibitors which included ROCK1/2 inhibitors such as Y27632 and GSK269962, the RhoA/B/C inhibitor C3-exoenzyme (tat-C3). Since the activity of several RhoGTPases upregulated during drug tolerance (Fig. 3c, Supplementary Fig. 11B and C) depends on their prenylation status[52], we also included a farnesyltransferase inhibitor (FTI, tipifarnib) and a geranylgeranyltransferase inhibitor (GGTi, GGTi-298). Inhibitors alone did not affect cell proliferation, except for the HCC4006 cell line in which tipifarnib displayed a cytostatic activity (Supplementary Fig. 12A). Interestingly, all these inhibitors interfered with the normal course of the adaptive response to EGFR-TKI, although they showed different activity profiles. For instance, ROCK inhibitors impaired osimertinib-induced MLC2 activation and stress fiber formation (Supplementary Fig. 12B-C) but failed to eliminate DTC and rather promoted a long-term stabilization of the G1 population (Fig. 3d, Supplementary Fig. 12A and D), together with an increased expression of the AT1-specific marker AGER (Supplementary Fig. 12B). This suggests that ROCK activation may be required for mesenchymal transition but is dispensable for AT1 non-cycling cell survival. RhoA/B/C inhibitor tat-C3 also decreased stress fiber formation (Supplementary Fig. 12C) and strongly increased EGFR-TKI efficiency, however some resistant clones could still be observed depending on the cell type (Fig. 3d, Supplementary Fig. 12A). Interestingly, amongst all the inhibitors tested, the FTI tipifarnib was the only drug that completely prevented the development of resistances to both osimertinib and erlotinib in all the cell lines tested (Fig. 3d and e, Supplementary Fig. 12A). Remarkably, tipifarnib also prevented relapse to targeted therapies in other models of oncogenic addiction such as ALK[EML4] and KRAS[G12] NSCLC or BRAF[V600E]-mutant melanoma cell lines treated with lorlatinib, sotorasib and dabrafenib, respectively (Fig. 3e, Supplementary Fig. 13A). Other FTIs such as CP-609754 or lonafarnib similarly prevented relapse to osimertinib, confirming that the effect was class-wide and not limited to a single drug (Supplementary Fig. 13B). Tipifarnib also induced cell death when administered after early relapse (Supplementary Fig. 13C), while fully resistant proliferative clones displayed different degrees of sensitivity to tipifarnib with some showing an almost complete response and some others a complete lack of sensitivity (Supplementary Fig. 13D). These results suggest that tipifarnib preferentially interferes with the earliest stages of drug adaptation, although it may retain some activity in fully resistant proliferative cells depending on the clone.

We next aimed to determine which farnesylated protein(s) could mediate both cell survival and phenotypic reorganization of tumor cells in response to TT. We first assessed mRNA expression levels of the genes coding for farnesylated proteins (40 according to a recent study[53]). Our scRNAseq data showed that while farnesylated GTPases such as RND3 (RHOE), RHOB, or RHOBTB3 were overexpressed by osimertinib during G1 arrest, the expression of several cell-division-related genes whose protein products are farnesylation-dependent, such as lamins (LMNB1, LMNB2, LMNA), centromere protein F (CENPF) or spindle apparatus coiled-coil protein 1 (SPDL1) was highly repressed during G1 but fully restored in S/G2 (Fig. 3f, left). This pattern was highly conserved amongst several models of DTC, suggesting that this balance of expression among farnesylated proteins could be a hallmark of drug tolerance (Fig. 3f, right). However, although siRNA-mediated inhibition of individual farnesylated proteins such as RHOB, RHOE, LMNB1, CENPF or HRAS[54] significantly increased sensitivity to TT, none could fully recapitulate tipifarnib ability to present relapse (Fig. 3g), which could suggest a multi-target mode of action for FTIs. To exclude potential off-target effects of tipifarnib, we knocked out FNTB, which codes for the beta subunit of farnesyltransferase, in PC9 cells (Supplementary Fig. 14A). FNTB depletion strongly prevented protein farnesylation as determined by a characteristic shift of the

HRAS protein (Supplementary Fig. 14A–C) similar to the shift observed in response to tipifarnib (Supp Fig. 14D). Strikingly, in vivo osimertinib treatment of mice bearing xenograft tumors from FNTB-KO PC9 cells led to a durable response for the entire 7.5-month treatment period (12/13, 92.3%), while most FNTB-WT PC9 tumors (8/11, 72.7%) had relapsed within this timeframe (Fig. 3h, i). Importantly, FNTB depletion did not affect tumor growth in vehicle-treated mice, indicating that the effect was specific to osimertinib treatment (Fig. 3h). Similar results were obtained in vitro with both FNTB-KO and FNTB-KD PC9 cells, suggesting that partial inhibition of the farnesyltransferase was sufficient to alter the emergence of osimertinib-resistant cells (Supplementary Fig. 14E).

Finally, we aimed to test the ability of tipifarnib to prevent TT-induced stress-fiber formation, and we observed divergent effects amongst the models (Supplementary Fig. 15A). In Calu-1 KRAS[G12C]-mutant cell line, tipifarnib strongly prevented sotorasib-induced MLC2 phosphorylation and stress-fiber formation (Supplementary Fig. 15A–C), and strongly prevented RhoE farnesylation as determined by a characteristic shift (Supplementary Fig. 15B). In this model, siRNA-mediated downregulation of RHOE strongly prevented pMLC2 and stress-fiber formation (Supplementary Fig. 15D–E), and also increased sensitivity to sotorasib (Supplementary Fig. 15F). Nonetheless, although RHOE downregulation could sensitize tumor cells in other oncogenic settings such as osimertinib-treated HCC4006 cells (Fig. 3g), its ability to modulate cytoskeleton reorganization was either moderated or not observed in other models, suggesting that other proteins may promote stress-fiber formation according to the cellular context.

Overall, we determined that drug tolerance is invariably associated with Rho/ROCK-mediated stress-fiber formation, although this phenotypic characteristic was not critical for DTC survival as highlighted by co-treatment with ROCK inhibitors, but might be required for cell re-proliferation. We also identified tipifarnib as a highly efficient drug in preventing relapse to a broad range of TT, although its efficacy does not seem to depend on its ability to modulate ROCK activity, but more likely involves the inhibition of multiple farnesylated proteins that display a highly conserved pattern of regulation among the drug-tolerant models.

## Osimertinib-tipifarnib (OT) co-treatment induces mitotic defects and ISR-mediated apoptotic pathway

We first aimed to decipher the physiological consequence of OT treatment on DTC. Time-lapse imaging revealed that although tipifarnib did not prevent the emergence of osimertinib-derived early escapers, escaping cells failed to undergo mitosis and ultimately died (Fig. 4a), suggesting that OT treatment may interfere with cell division, which was consistent with an inhibition of cell-cycle related proteins such as lamins, CENPF or RhoGTPases, amongst others (Fig. 3f). We also observed that a high percentage of OT-treated cells that progressed to S/G2 returned to G1 without dividing, consistent with a process of mitotic slippage or endoreplication[55,56], which was also observable at lower frequency in osimertinib-treated cells (Fig. 4a, Supplementary Fig. 16, Supplementary Movie 5).

To decipher the molecular mechanisms underlying the ability of tipifarnib to prevent relapse to osimertinib, we performed scRNAseq on OT-treated G1 and S/G2 sorted cells. OT-treated cells displayed a distinct transcriptomic profile compared to osimertinib-treated cells (Fig. 4b, Supplementary Fig. 17A). Strikingly, co-treatment with tipifarnib strongly prevented osimertinib-induced overexpression of most genes associated with drug-tolerance, such as SPARC, IGFBP5, ANOS1, CRYAB or AXL (Fig. 4b), resulting in a significant downregulation of DTC_UP and DTC_DOWN signatures in both G1 and S/G2-treated cells (Fig. 4c). Indeed, OT treatment strongly interfered with RHO_GTPASE_CYCLE, AT1 and mesenchymal-related signatures (Fig. 4d, e, Supplementary Fig. 17B). Consistently, interferon and EMT signatures were

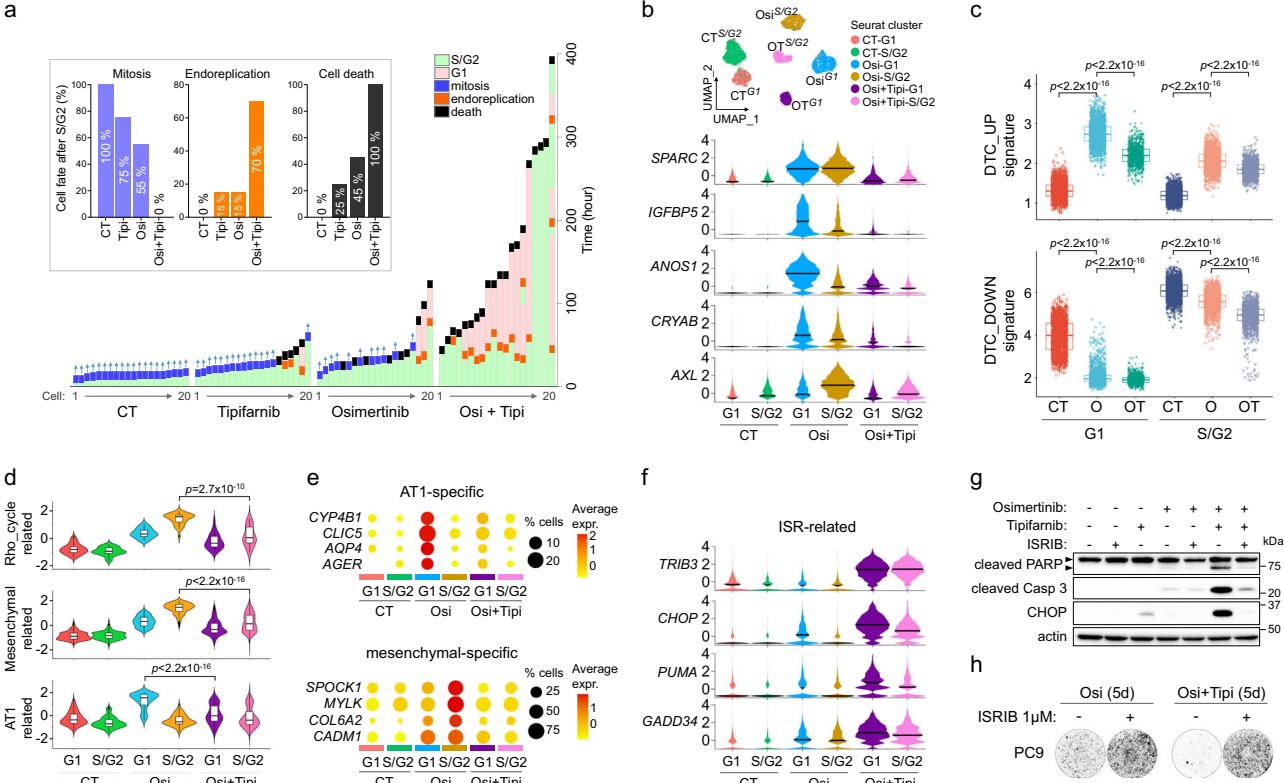

**Fig. 4 | Tipifarnib alters the adaptive response to osimertinib and induces ISR-mediated apoptotic pathway. a** Cell fate after progression to S/G2 of HCC4006 subclones treated or not with osimertinib (1 μM), tipifarnib (1 μM) or the combination (Osi+Tipi). Bottom: The data represents the monitoring of cells (n = 20/condition) since their entry to S/G2 phase, with the time spent for each phase/event of the cell cycle. Top: The bar plot recapitulates the percentage of cells that experienced mitosis and endoreplication after S/G2 phase, and the percentage of cells that died before dividing. **b** *Top*: UMAP plot of the different clusters of untreated, osimertinib-treated and osimertinib+tipifarnib-treated G1 and S/G2 HCC4006 subclones obtained by scRNAseq. *Bottom*: Distribution of normalized expression levels (z-score) of DTC-related genes regulated by tipifarnib-osimertinib co-treatment. **c** Signature scores of DTC_UP and DTC_DOWN in untreated (CT), osimertinib (O) and osimertinib + tipifarnib (OT) treated G1 and S/G2 cells. The box plots display 25th (lower bound), 50th (center, median), and 75th (upper bound) percentiles, with whiskers generated with the Tukey method; all points are shown. *p*-value was calculated using a two-sided Wilcoxon test. **d** Signature scores of RHO_GTPase_cycle-related, mesenchymal-related and alveolar type 1 (AT1)-related gene signatures in untreated (CT), osimertinib (Osi) and osimertinib + tipifarnib (Osi+Tipi) treated G1 and S/G2 cells. The box plots within violin plots are similar to c. *p*-value was calculated using two-sided Wilcoxon test. For **c** and **d**: CT-G1, n = 1678; O-G1, n = 2097, OT-G1, n = 897; CT-SG2, n = 2654; O-SG2, n = 1963; OT-SG2, n = 1266. **e** Dot plot showing the expression level and percentage of cells expressing genes specific for the alveolar and mesenchymal phenotypes in the different clusters. **f** Distribution of normalized expression levels (z-score) of genes related to integrated stress response (ISR) pathway. **g** Western blot analysis of proteins related to apoptosis (PARP and caspase-3) and ISR (CHOP) of HCC4006 subclones treated with osimertinib (1 μM), tipifarnib (1 μM) and ISR inhibitor (ISRIB, 1 μM) alone or in combination. Representative blots from n = 3 independent biological experiments. The upper arrow shows the total and the lower arrow shows cleaved PARP. **h** Crystal violet staining of PC9 cells pre-treated or not with ISRIB (1 μM, 24 h) and treated for 5 days with 1 μM osimertinib alone or in combination with 1 μM tipifarnib. Representative images from n = 3 independent biological experiments. Source data are provided as a Source data file.

the most significantly downregulated pathways in G1 and S/G2 OT-treated cells, respectively (Supplementary Fig. 17C–E). Finally, we sought to determine the molecular mechanisms responsible for cell death under OT co-treatment. We observed a strong enrichment in unfolded protein response (UPR)/integrated stress response (ISR)-related gene signatures in both G1 and S/G2 OT-treated cells (Supplementary Fig. 18A, B), which was concordant with a strong over-expression of ATF4-regulated genes such as *DDIT3/CHOP*, *TRIB3*, *CHAC1*, *PSAT1*, *FAM129A/NIBAN1*, *PPP1R15A/GADD34* or the pro-apoptotic *BBC3/PUMA*, among others (Fig. 4f). Upregulation of ATF4 and CHOP by OT treatment was confirmed at the protein level and correlated with PARP and Caspase 3 cleavage (Supplementary Fig. 18C). Most importantly, pharmacological inhibition of ISR using ISRIB strongly prevented cell death (Fig. 4g, h), confirming that activation of this pathway was responsible for OT-induced cell death. Interestingly, inhibition of ISR also prevented sotorasib+tipifarnib-induced death in KRAS^G12C models (H23 and Calu-1) and lorlatinib +tipifarnib in the ALK^EML4 model H3122, suggesting a common mode of

action of tipifarnib in other oncogenic settings (Supplementary Fig. 18D).

Overall, we observed that co-treatment with tipifarnib induces lethal mitotic defects when combined with osimertinib, and broadly interferes with the drug-tolerant state. OT treatment resulted in the activation of the ATF4/CHOP stress response pathway, which resulted in the apoptosis of co-treated cells.

**Tipifarnib prevents relapse to targeted therapies in vivo**
Lastly, we aimed to translate our findings to pre-clinical in vivo models by testing the combination of tipifarnib with different targeted therapies. We first performed osimertinib and OT treatments in two different EGFR-mutant NSCLC PDX models harboring respectively an EGFR^L858R/T790M double mutation (model TP103[57]) and an exon 20 insertion (model LU0387[58]), as well as in a PC9 xenograft model. For all three models, OT co-treatment showed better anti-tumor efficacy than osimertinib alone, and most importantly, the addition of tipifarnib strongly and durably prevented the emergence

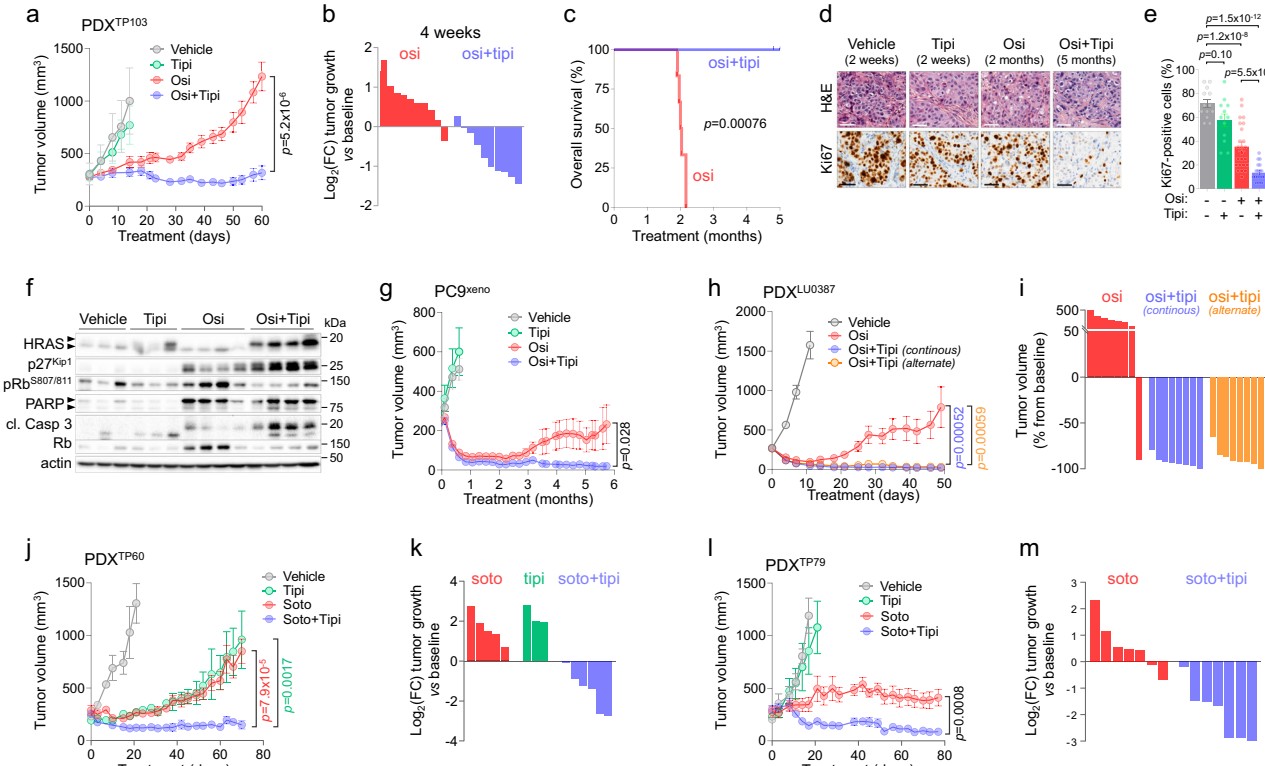

**Fig. 5 | Tipifarnib prevents relapse to targeted therapies in vivo. a** Mean tumor volume of the TP103 NSCLC PDX model (EGFR$^{L858R/T790M}$) treated 5 days/week with vehicle ($n = 5$), tipifarnib (tipi, 80 mg/kg, b.i.d., $n = 5$), osimertinib (osi, 5 mg/kg, q.d, $n = 10$) or by the combination (osi+tipi, $n = 10$). Data are mean ± SEM; $p$-value was calculated using two-tailed unpaired $t$-test. **b** Log2FC tumor growth $vs$ baseline at 4-week treatment with osimertinib or osimertinib+tipifarnib. **c** Overall survival of the mice treated with osimertinib or osimertinib+tipifarnib. The graph is the result of one cohort of mice with $n = 6$ mice in both arms. $p$-value was calculated using the log-rank Mantel-Cox test. **d** Representative images of Hematoxylin and Eosin (H&E) staining and Ki67 IHC from PDX tumors collected after indicated times and treatments. Data are representative of $n = 3$ (vehicle, tipifarnib), $n = 6$ (osimertinib) and $n = 5$ (osimertinib+tipifarnib) independent tumors. Scale bar: 50 μm. **e** Ki67 IHC scores quantified from 4 different zones of each independent tumor presented in **d**. **f** Western Blot analysis of individual PDX after 2 weeks (vehicle, $n = 3$; tipifarnib, $n = 3$), 2 months (osimertinib, $n = 4$) and 5 months (osi+tipi, $n = 4$) treatment. For HRAS: the upper arrow shows unfarnesylated and the lower arrow shows farnesylated protein; for PARP: the upper arrow shows total and the lower arrow shows cleaved protein. **g** Mean tumor volume of PC9 xenografts treated 5 days/week with vehicle ($n = 6$), tipifarnib (Tipi, 80 mg/kg, b.i.d., $n = 6$), osimertinib (osi, 5 mg/kg,

q.d, $n = 10$), or by the combination (osi+tipi, $n = 12$). Data are mean ± SEM; $p$-value was calculated using two-tailed unpaired $t$-test. **h** Mean tumor volume of the LU0387 NSCLC PDX model (EGFR$^{exon20}$ insertion) treated 5 days/week with vehicle ($n = 8$), osimertinib (osi, 25 mg/kg, q.d, $n = 8$), or the combination (osimertinib + tipifarnib at 60 mg/kg, continuously b.i.d, $n = 8$, or intermittently 1 week ON/1 week OFF, $n = 8$). Data are mean ± SEM; $p$-value was calculated using two-tailed unpaired $t$-test. $p$-value (osi+tipi continuous $vs$ osi) is shown in blue and (osi+tipi alternate $vs$ osi) in orange. **i** Percentage of LU0387 tumor volume $vs$ baseline at day 49. **j** Mean tumor volume of the TP60 NSCLC PDX model (KRAS$^{G12C}$) treated with vehicle ($n = 3$), tipifarnib (tipi, 80 mg/kg, b.i.d., $n = 3$), sotorasib (soto, 30 mg/kg, q.d, $n = 5$), or the combination (soto+tipi, $n = 6$). Data are mean ± SEM; $p$-value was calculated using two-tailed unpaired $t$-test. $p$-value (soto+tipi $vs$ soto) is shown in red and (soto +tipi $vs$ tipi) in green. **k** Log2 fold change of the TP60 PDX tumor size compared to baseline at day 70. **l** Mean tumor volume of the TP79 NSCLC PDX model (KRAS$^{G12C}$) treated with vehicle ($n = 6$), tipifarnib (tipi, 80 mg/kg, b.i.d., $n = 4$), sotorasib (soto, 30 mg/kg, q.d, $n = 7$), or the combination (soto+tipi, $n = 7$). Data are mean ± SEM; $p$-value was calculated using two-tailed unpaired $t$-test. **m** Log2 fold change of the TP79 PDX tumor size compared to baseline at day 77. Source data are provided as a Source data file.

of osimertinib resistance, with no evidence of toxicity as shown by stable body weight and good general aspect of mice (Fig. 5, Supplementary Fig. 19B, F, G and 20D, F, G). In the TP103 model, osimertinib induced a moderate response and relapse was observed for all (10/10) mice within 2 months, whereas the addition of tipifarnib induced significantly higher tumor regression, with an almost stable disease lasting the entire dosing period of 5 months (Fig. 5a–c, Supplementary Fig. 19A, D, E, H). Consistent with in vitro data, OT-treated tumors showed a strongly reduced proliferative state even after 5 months of treatment as highlighted by Ki67 staining (Fig. 5d, e). This correlated with increased p27$^{Kip1}$ levels and reduced phospho-Rb, as well as increased PARP and Caspase 3 cleavage, suggesting that non-cycling OT-treated cells were undergoing apoptosis (Fig. 5f). Tipifarnib-mediated farnesyltransferase inhibition was highlighted by a characteristic shift of the HRAS protein toward a non-farnesylated state (Fig. 5f). Similar results were obtained in the PC9 xenograft model, where OT treatment induced a strong, stable and

apparently safe response for up to 6 months, with almost complete tumor regression in all animals, whereas osimertinib induced a less potent effect and relapse was observed in 4/10 tumors (Fig. 5g, Supplementary Fig. 20A–D). For the LU0387 model, tipifarnib was administered either continuously or intermittently (1 week on, 1 week off), to better reflect the treatment regimen generally used in clinical practice for this molecule[54]. Despite a strong initial anti-tumor effect of osimertinib, 7/8 (87.5%) tumors relapsed within 20 days of treatment, whereas both continuous and intermittent OT treatments induced stronger tumor regression and a stable response in 16/16 tumors (100%) for the entire 50-days treatment period (Fig. 5h and i, Supplementary Fig. 20E). Notably, tipifarnib alone did not display anti-tumor activity when used alone in the TP103 and PC9 xenograft models (and was not assessed in LU0387 PDX), highlighting a synthetic lethality between osimertinib and tipifarnib in the context of the adaptive response to TT (Fig. 5a and g, Supplementary Fig. 19A, D, E, H and 20C).

Based on our in vitro results in KRAS[G12C]-mutant models, we aimed to evaluate the combination of tipifarnib with sotorasib in two different KRAS[G12C]-mutant NSCLC PDX models (TP60 and TP79)[59]. Strikingly, combinatory treatment induced a much stronger and durable antitumor response in both models compared to sotorasib alone (Fig. 5j–m), with a stable response during the entire dosing period of 70 days for TP60 (Fig. 5j) and 77 days for TP79 (Fig. 5l). As with the combination with osimertinib, the tipifarnib and sotorasib pairing was well-tolerated as indicated by stable body weight and good general aspect of mice (Supplementary Fig. 20F, G).

Altogether, our in vivo data show that tipifarnib safely and durably prevents the emergence of resistances to osimertinib in EGFR-mutant and to sotorasib in KRAS[G12C]-mutant NSCLC, thus providing a strong rationale to evaluate those combinations in the clinic.

## Discussion

Since the discovery two decades ago that single-agent targeted therapy could induce dramatic clinical responses in oncogene-addicted tumors (i.e., gefitinib in EGFR-mutant NSCLC)[60,61], the constant and growing identification of oncogenic drivers and subsequent development of corresponding TT have fundamentally reshaped the standard of care for more than half of NSCLC patients[18]. Nevertheless, the inevitable emergence of resistance remains to date a major public health issue for all TT and across all types of cancer that has not been solved despite the improvements of successive generations of inhibitors. Yet, this therapeutic approach remains the best option for oncogene-addicted NSCLC patients, prompting the search for new strategies to prevent the emergence of resistance.

In this context, the identification of DTCs[20] has emerged as a concept that could shed light on the very first steps of drug resistance. The paradigm of drug adaptation was exemplified by Ramirez et al. in 2016, when the authors determined that different genetic resistance mechanisms could emerge from a genetically homogeneous population of clonally-derived EGFR-mutant PC9 cells in response to erlotinib[21]. While it is widely accepted to define DTCs as dormant or slow-to-non cycling cells, a surprisingly substantial gap remains in our understanding of the dynamics and molecular processes underlying their entry into and exit from dormancy.

To address this pivotal question, we applied the FUCCI system[43] in a panel of clonal EGFR-mutant cell lines, which enabled accurate monitoring of the cell cycle dynamics throughout the adaptive response to EGFR-TKIs. This strategy offered several significant advantages: firstly, it allowed the identification and characterization of a rare fraction of proliferating cells that emerged shortly after treatment initiation among a stably and progressively dying G1-arrested population. To our knowledge, these early escapers represent the earliest population of resistance-initiating cells isolated so far. These cells displayed a mesenchymal-like phenotype and expressed known markers of resistance such as SERPINE1 or AXL[26], together with enrichment in RHO_GTPASE_CYCLE, actin cytoskeleton-related, NRF2 and glutathione metabolism-related gene signatures. Early escapers exhibited traits resembling "cycling persisters", a cell population recently identified by Oren using the Watermelon system[35]. Our method offers the substantial advantage of identifying those cells prior to molecular analysis, thereby facilitating their live collection, enrichment, and subsequent analysis. Secondly, one of the most interesting findings arose from the scRNAseq analysis of the osimertinib-treated G1 population, which revealed two distinct clusters: the first one was strongly enriched in AT1-specific genes such as AGER, AQP4 or CLIC5, and the second one was enriched in mesenchymal-related genes such as SERPINE1, MATN2 or SPOCK1. Strikingly, an AT1 signature was recently identified by T. Bivona's lab in residual disease (RD) from NSCLC patients treated with osimertinib, while plasminogen activator inhibitor type 1 (SERPINE1) was amongst the top overexpressed genes in progressive disease (PD) compared to RD[62], although the relationship between both phenotypes was not assessed in that study. Here, we performed pseudotime analysis which revealed a tight connection between the alveolar (AGER-positive) and mesenchymal (SERPINE1-positive) osimertinib-treated G1 clusters, suggesting that early escapers may have emerged from previously AT1-like cells. Moreover, the AT1 signature was one of the unique common features shared by all the models of drug-tolerance, together with increased contractile signatures, which were also highly upregulated in healthy tissue compared to lung adenocarcinoma, reinforcing the evidence that drug-tolerant cells shared phenotypic similarities with healthy lung cells. Although our study cannot exclude that AT1-like and mesenchymal cells could have undergone distinct paths of drug adaptation at some point during the adaptive response, the timing of emergence of both populations and their tight phenotypic similarities revealed by scRNAseq suggest that the alveolar-like population may act as a reservoir from which mesenchymal cells could emerge. Consistent with this hypothesis, a recent study from T. Tammela's lab highlighted the crucial role of AT1 differentiation in the emergence of resistance to KRAS-inhibitors in NSCLC[63], strongly supporting our findings. Interestingly, a direct link between myosin activation, actin cytoskeleton remodeling and AT1 cell fate, which were the unique common characteristics across all the DTC models, have been recently reported by Edward E. Morrisey's lab[64], supporting an interconnection between these phenotypic features.

We then asked whether these phenotypic hallmarks could also represent a common vulnerability of DTCs. While ROCK-mediated cytoskeletal reorganization systematically accompanied the drug-tolerant phenotype, this was not required for DTC survival as determined using ROCK-inhibitors co-treatments but seemed to be rather necessary for cell re-proliferation. Interestingly, ROCK-myosin II pathway and cytoskeletal remodeling have recently been shown to drive resistance to TT in BRAF-mutant melanoma, and a combination of ROCK inhibitors (ROCKi) and BRAF inhibitors (BRAFi) induced regression of BRAFi-resistant tumors in vivo[65]. In our models, ROCKi robustly impaired the emergence of resistances to TT but failed to fully eradicate the reservoir of DTC, which suggests a different degree of dependency of ROCK activity in DTC derived from NSCLC compared with melanoma.

Another major albeit unexpected finding of our study came from our drug screen which highlighted FTIs such as tipifarnib as the only compounds to completely prevent the development of resistances and eliminate the DTC population across all our models. We validated the specificity of the target by creating a FNTB-KO model, which accurately mimicked the effects of FTIs, thus eliminating a potential role for off-target effects. We observed a conserved expression pattern of farnesylated proteins across the models of drug tolerance which invariably involved overexpression of Rho-GTPases (RHOB, RHOE/RND3, RHOBTB3) and downregulation of centromere protein F (CENPF) and lamins (LMNA, LMNB1, LMNB2) that could favor a global phenotypic dependency on farnesyltransferase activity. Although the downregulation of each of these proteins individually increased sensitivity to TT, none could fully recapitulate FTI's ability to prevent resistances, ruling out the possibility that a unique farnesylated protein could mediate the effect and rather pointing to a multi-target mechanism of action. Although we might have expected tipifarnib to prevent the emergence of TT-induced actin stress fibers by targeting RhoGTPases in the same way as C3-exoenzyme, this ability appeared to depend on the cellular model, thus disconnecting the capacity of tipifarnib to prevent resistances and its action on the cytoskeleton. Yet, we could highlight an unexpected role for RhoE, which usually acts as a negative regulator of ROCK-mediated actomyosin contractility[66], but was found to be the farnesylated protein responsible for stress fiber formation in response to sotorasib in the KRAS[G12C]-mutant Calu-1 model, in which tipifarnib showed the greatest efficacy in preventing cytoskeletal reorganization. Unlike in other models of DTC where several other

Rho-GTPases (either farnesylated or not) were overexpressed in response to TT and could potentially promote cytoskeletal rearrangement, RhoE was the only one overexpressed in response to sotorasib in the Calu-1 model, which could explain the different phenotypic responses to FTI across the models. Besides its role in regulating stress fibers, RhoE could also act on both cell survival/apoptosis or cell cycle progression through a ROCK-independent mechanism[67].

One of the most striking phenotypic impacts of the combination of TT and FTI was observed using the FUCCI system, which revealed the ability of this combination to completely prevent mitosis of early escapers, ultimately resulting in cell death. In addition to a potential role of Rho-GTPases such as RhoE in the cell division process[67], prevention of farnesylation of proteins such as centromere proteins[68,69] and lamins[70], whose expression was strongly repressed in G1- but fully restored in S/G2-treated cells, may also be responsible for impaired mitosis of early escapers. Targeting lamins with FTIs, especially lamin B1, could also have broader repercussions on gene expression, as nuclear lamina interacts with and regulates genome expression through regions called lamina associated domains (LADs)[71]. Interestingly, Morrisey's study[64] revealed that alveolar reprogramming was highly dependent on genome organization through LADs, in which lamin B1 played a major role in determining AT1-specific gene expression. It is therefore not to exclude that prevention of lamin B1 farnesylation by FTI may influence AT1 phenotypic reprogramming during TT treatment.

We found that FTI and TT co-treatment induced cell death by inducing the ATF4/CHOP-mediated apoptotic pathway, which could be prevented by the ISR inhibitor ISRIB in all the models of oncogenic addiction tested. ATF4/CHOP-mediated apoptosis may be the result of the activation of multiple pathways such as the ISR[72] or the UPR, which senses the accumulation of misfolded proteins in the endoplasmic reticulum (ER) through PERK (PKR-like ER kinase)[72]. In the latter scenario, the alteration in the expression of multiple farnesylated proteins observed in G1-arrested cells but also when cells regained proliferative capacities could lead to proteotoxic stress due to conformational destabilization of those proteins caused by the absence of their farnesyl group. ISR can also be triggered by other kinases such as PKR (double-stranded RNA-dependent protein kinase), HRI (heme-regulated eiF2a kinase) and GCN2 (general control nonderepressible 2), depending on the environmental and physiological stress (reviewed in Pakos-Zebrucka et al.[73]). We cannot exclude atypical activation of ATF4 as it has been described in mantle cell lymphoma cell lines, where ATF4 was overexpressed independently of the four main kinases activation[74].

Finally, our study highlighted several major inputs regarding clinical perspectives. We provide a signature of drug tolerance, which comprised the 212 most commonly upregulated and 495 most commonly downregulated genes across the different models. This signature should be useful for the detection and characterization of minimal residual disease in vivo and in patients, as well as for the identification of potential vulnerabilities of DTCs. We also revealed the remarkable efficacy of the in vivo combination of the FTI tipifarnib with targeted therapies including osimertinib and sotorasib. Tipifarnib was the first selective FTI to enter clinical trials over two decades ago, and is well-tolerated and clinically active, especially in *HRAS*-mutant head and neck squamous cell carcinoma (HNSCC) patients[54]. Interestingly, a good tolerability profile of tipifarnib and erlotinib combination has already been reported in an early phase I study, although the evaluation of the combination's efficacy was not relevant because the study was performed in unselected patients (i.e., EGFR WT tumors) who are not sensitive to EGFR-TKI[75].

There are some limitations in this study. First, the relationship between lineage plasticity during the adaptive response to TT and the vulnerability to farnesyltransferase inhibition is still unknown. This mechanistic gap between the hallmarks of drug tolerance and the FTIs effects partially stems from the lack of formal identification of the farnesylated proteins involved in the phenotypic reprogramming and responsible for the relapse. Second, the mechanism by which FTI treatment induces the integrated stress response has not been elucidated in this work and should be further investigated. Finally, the precise therapeutic window for the co-treatment of targeted therapies and FTIs, and the specificity of the FTI effect for the drug-tolerant state, remain to be clarified. This is probably due to a lack of a clear definition of the drug-tolerant state, particularly defining when drug tolerance begins and when resistance takes over. Additional experiments are required to understand why some early relapsing or fully resistant proliferative cells retain sensitivity to FTI, and why some others do not.

In summary, we provide an approach to address the molecular complexity underlying the dynamic plasticity of tumor cells during the early steps of resistance to TT, across different models of oncogenic addiction. We identified AT1-like differentiation as one of the earliest events in the adaptive response to TT in NSCLC, which was consistently associated with deregulation of farnesylated protein expression. These findings pave the way for the evaluation of treatments combining TT and FTI as a way to effectively prevent tumor relapse and ultimately lead to prolonged treatment response in oncogene-addicted NSCLC patients.

## Methods

### Ethics
This study complies with all relevant ethical regulations. Animal care complied with European and national legislation guidelines for the use of laboratory animals and experimental procedures were approved by the CREFRE ethical committee (APAFIS #31739-2021051816536458).

### Cell culture
The human NSCLC cell lines HCC4006 (CRL-2871, EGFR$^{\Delta L747-E749}$, A750P), HCC827 (CRL-2868, EGFR$^{\Delta E749-A750}$), HCC2935 (CRL-2869, EGFR$^{\Delta E746-T751}$, S752I), Calu-1 (HTB-54, KRAS$^{G12C}$), H23 (CRL-5800, KRAS$^{G12C}$), and H3122 (EML4-ALK rearrangement) cell lines were obtained from the American Type Culture Collection (Manassas, VA, USA). The H3255 NSCLC cell line (EGFR$^{L858R}$), the PC9 NSCLC cell line (EGFR$^{del\ E746-A750}$), the A375 cell line (CRL-1619, BRAF$^{V600E}$), and the HCC364 BRAF$^{V600E}$ cell line, were a kind gift from Helene Blons (APHP, Paris, France), Antonio Maraver (IRCM, Montpellier, France), Nathalie Andrieu (CRCT, Toulouse, France), and David Santamaría (CIC, Salamanca, Spain), respectively. NSCLC cell lines and A375 melanoma cell line were cultured in RPMI (Roswell Park Memorial Institute) 1640 and DMEM medium (Dubelcco's Modified Eagle Medium) respectively, containing 10% fetal bovine serum (FBS), and were maintained at 37 °C in a humidified chamber containing 5% CO2. Cell lines were authenticated using short tandem repeat DNA profiling and tested for mycoplasma contamination within the experimental time frame. For each EGFR-mutant cell line, subclones were generated by limiting dilution. Moreover, subcultures were conducted for a limited period of time in order to limit cell deviation. Cells were imaged with a ZEISS Axio Vert.A1 microscope and analysed with ZEN software (ZEISS).

### Cell cycle analysis
Cells were transduced with Incucyte® Cell Cycle Green/Red Lentivirus Reagent (EF1α-Puro) (Sartorius, Cat. No. 4779) as recommended by the manufacturer and transduced cells were selected by puromycin treatment at 1 μg/mL for one week. Cells were imaged every hour with Incucyte® S3 Live-Cell Analysis System (Sartorius). G1-phase, S/G2-phase and G1/S-phase cells were quantified using Incucyte® S3 Live-Cell Analysis 2020B software (Sartorius). Single-cell tracking and quantification of nuclear size were made using Incucyte-based images.

### Cytotoxicity assay
The viability of untreated cells or treated cells for 5 days was assessed using CellTiter 96® AQueous One Solution Cell Proliferation Assay

(MTS) (Promega®, ref:G3580) according to the manufacturer's instructions.

## Western blot analyses

For in vitro experiments, cells were lysed with RIPA buffer (Tris-HCl 50 mM, NaCl 150 mM, Triton X-100 1%, EDTA 5 mM, Sodium Deoxycholate 0.5%, SDS 0.1%) complemented with proteases- and phosphatases-inhibitors. For animal experiments, frozen tumors were ground and lysed with Tris-SDS 1% buffer complemented with proteases- and phosphatases-inhibitors; lysates were sonicated. Protein content was quantified using the Bradford method. Protein extracts were separated on SDS-PAGE and electrotransferred on polyvinylidene difluoride membranes. Blots were probed with primary antibodies reported in Supplementary Table 3. Detection was performed using peroxidase-conjugated secondary antibodies and a chemiluminescent detection kit (Clarity™ Western ECL, Bio-Rad) with a ChemiDoc™ MP Imaging system (Bio-Rad) and images were retrieved using Image Lab 6.1 software. Reference of antibodies and dilutions are shown in Supplementary Table 3.

## Cell sorting

Cells transduced with the FUCCI system[43] were dissociated with trypsin, recovered in FACS buffer (0.04% BSA in PBS) and kept on ice. G1 (red) and S/G2 (green) cells were sorted at 4 °C using FACS Melody and FACSChorus v1.3.3 software (BD Biosciences). Sorted cells were then processed differently according to the downstream analysis. For Western Blot analyses, cells were recovered in PBS, spun down and lysed in RIPA buffer. For proliferation analysis, cells were recovered in a cell culture medium and 5.000 cells were plated per well in a 96-well plate with osimertinib 1 μM.

## Single-cell RNA-sequencing

HCC4006 subcloned cells were expanded in untreated, osimertinib- or osimertinib + tipifarnib-containing medium for 4, 20 and 16 days respectively. The medium was changed 24 h before sorting. Cells were dissociated by trypsinisation, recovered in FACS buffer (0.04% BSA in PBS) and kept on ice. G1 (red) and S/G2 (green) cells were sorted at 4 °C using FACS Melody (BD Biosciences). Cells were spun down and resuspended in appropriate volume for a final concentration of 500 to 1500 cells / μl, with a viability above 85%. 6000 cells per sample were loaded to the Chromium Controller (10X Genomics) and 1900-3300 cells were recovered depending on the sample. scRNA-seq libraries were generated using the 10X Genomics Chromium Single Cell 3' Kit v2 according to manufacturer instructions. The libraries were profiled with the HS NGS kit for the Fragment Analyzer (Agilent Technologies) and quantified using the KAPA library quantification kit (Roche Diagnostics). The libraries were pooled and sequenced on the Illumina NextSeq550 instrument by the CRCT Genomics platform using the High Output 150 cycles kit. The single-indexed sequencing parameters were 28, 8, 0, 91 cycles (read 1, index 1, index 2, read 2). The dual-indexed sequencing parameters were 28, 10, 10, 90 cycles (read 1, index 1, index 2, read 2). An average depth of ~46000 reads/cell was obtained.

After sequencing, all BCL files were converted to FASTQ files, and then these together with the human transcriptome of reference were used for alignment and raw count matrix generation, using cellranger 6.0.0 software (10X Genomics). Then, a quality control check was performed for each sample using Seurat v4.0 (R package), and retained cells from the different conditions were pulled together to create a global counts matrix. First, a linear and non-linear dimension reduction, principal component analysis (PCA) and a Uniform Manifold Approximation and Projection (UMAP) were performed. Next, a non-supervised clustering was performed using the FindClusters function from Seurat (R package). The differential gene expression analyses were done using FindMarkers or FindAllmarkers functions

from the same library within R environment. The p-value threshold was set by default at 5%. The gene enrichment analyses were performed using a SingleCell Signature Explorer (https://doi.org/10.1093/nar/gkz601), and a resulting matrix of cells × signatures scores was obtained. The evaluated signatures were obtained from MsigDB (https://www.gsea-msigdb.org/gsea/msigdb), unless otherwise stated. Violin plots of the mean pathway scores were computed and displayed using ggplot2 (R package). The heatmaps were generated using the ComplexHeatmap (R package). Trajectory differentiation inference was performed using Dynverse library with the R environment, which takes as an input the normalized and non-normalized expression matrix. Next, RNA velocity was also applied to these datasets by scvelo v0.2.4[76] in python environment, data was extracted from R environment and prepared to be used as an input to this pipeline. Finally, the latent time was calculated using scVelo, based on the RNA velocity. Single cell score for geneset analysis was computed as described[77]. Briefly, Single-Cell Signature Scorer computes the score of cell $C_j$ for gene set $GS_x$ as the sum of all UMI for all the $GS_x$ genes expressed by $C_j$ divided by the sum of all UMI expressed by $C_j$: Score of cell $C_j$ for geneset $GS_x = (\sum(UMI)_{GSx})/(\sum(UMI)_{Cj})$[77]. Other softwares used in this study for scRNAseq analysis were: scanpy v1.7.2, anndata v0.7.6, loompy v2.0.16, SingleCellExperiment v1.12.0, scran v1.18.5, and edgeR v3.32.1.

## RNA sequencing

Subclones generated from HCC4006, PC9 and H3255 cell lines were treated with 1 μM erlotinib for 24 h, 21 days (DTC state). For resistant proliferative cells (RPC), individual early-emerging colonies were isolated after 7–15 days of treatments and cultured in the presence of the drug for more than 3 months. HCC827 were treated until reaching the drug-tolerant state (11 days) with 1 μM osimertinib. RNA extraction was performed using AllPrep DNA/RNA Mini kit (Qiagen, #80204) according to the manufacturer's protocol. RNA quality was assessed using Fragment Analyzer (Agilent technologies) and the RQN values were provided to confirm the integrity of total RNA. RNA concentration was determined by fluorescent method using Quant-iT™ RNA Assay Kit, Broad Range (ThermoFisher Scientific). RNA samples were processed with Illumina TruSeq® Stranded mRNA Library Preparation Kit following the manufacturer's protocol. Library size and quality were confirmed on the Fragment analyzer (Agilent Technologies). KAPA quantification kit for Illumina platforms (KAPA Biosystems, Roche) was used to quantify the library by qPCR. Indexed libraries were pooled and sequenced on an Illumina NextSeq 550 (2×75 bp paired-end reads).

Reads were mapped and counted using the RNA-Seq by Expectation Maximization (RSEM) software v. 1.3.3 (with bowtie2-2.3.5.1) based on the human reference genome UCSC hg38. Differential expression was analyzed with DESeq2 v1.34.0.

## TCGA and GSEA analysis

RSEM-normalized expression data of human lung adenocarcinomas (n = 58) and corresponding normal adjacent tissue (n = 58) from The Cancer Genome Atlas (TCGA) database were downloaded from http://firebrowse.org. Pathway Gene Set Enrichment Analysis (GSEA) was performed with GSEA v4.1.0 software (https://www.gsea-msigdb.org/gsea).

## Filamentous actin staining

Cells were washed with PBS, fixed with paraformaldehyde 4% for 10 min, permeabilized with a solution of BSA 0.1%, Triton X-100 0.5% in PBS, blocked with a solution of BSA 1%, Triton X-100 0.1% in PBS and stained using Alexa Fluor™ 594 phalloidin (ThermoFisher Scientific, #A12381) diluted 50 times in a PBS 1X, BSA 0.1%, Triton X-100 0.1% solution. Nuclei were stained with DAPI (4′, 6-diamidino-2-phenylindole, dihydrochloride, ThermoFisherScientific, #D1306). Images

were acquired with a ZEISS Axio Vert.A1 microscope and analysed using Image J 1.52p software.

## Crystal violet staining

Cells were washed with PBS, fixed with paraformaldehyde 4% for 10 min and stained for 10 min with a solution containing 0.5% crystal violet (Sigma-Aldrich/Merck, ref: C3886) and 25% methanol in PBS. After 3 washes and drying, cell staining was imaged with a ChemiDoc™ MP Imaging system (Bio-Rad).

## Inhibitors

Cells were treated with inhibitors at the indicated concentrations two to three days after seeding, with medium changes twice a week for the entire duration of the experiment. Reference and concentrations of drugs are listed in Supplementary table 4.

## CRISPR

FNTB KO PC9 cells have been generated as previously described in Mandegar et al. (Cell stem cell 18, 541-553, 2016). Briefly, PC9 cells were transfected with JetPrime following the supplier protocol with the CRISPRn knock in vector (Addgene 73500) and each AAVS1-TALEN pair (Addgene 59025 and 59026) and selected with puromycin at 1 μg/ml for 2 weeks. Three gRNA oligos were designed with CHOPCHOP (FNTBn1: CGGCAGATGCGATTTGAAGG, FNTBn2: CCGCGCTGGTAAT CCTCAAG, FNTBn5: CAACCGCTCTCGCGCGTGCT). The oligos were then phosphorylated, annealed, and cloned into the pgRNA-CKB vector (Addgene 73501) using the BsmBI ligation strategy. The gRNA-expression vector was transfected into the CRISPRn cells with JetPrime following the supplier protocol. Blasticidin selection at 5 μg/ml was applied for two weeks. Cas9 expression was induced by doxycycline at 1 μg/ml for 72 h and subcloning was performed by limiting dilution. Clones were expanded and FNTB protein expression was checked by Western Blot.

## SiRNA-mediated gene silencing

Twenty-four hours after seeding, cells were transfected using jet-PRIME® Polyplus transfection® reagent for 4 h with siRNA (SMARTpool Dharmacon™) at a final concentration of 10 nM, and 1 μM osimertinib was added 24 h hours after transfection. Transfection was repeated twice a week while maintaining osimertinib treatment. References used are: SiGENOME Human RHOB (388), SiGENOME Human RND3 (390), SiGENOME Human LMNB1 (4001), SiGENOME Human HRAS (3265), SiGENOME Human CENPF (1063), SiGENOME non-targeting siRNA pool#1.

## Animal models

All breeding, mouse husbandry, and in vivo experiments were performed with the approval of CREFRE ethical committee. All procedures involving animals and their care conformed to institutional guidelines for the use of animals in biomedical research. Animals were housed under controlled temperature and lighting (12/12-h light/dark cycle). Both males and females have been used for in vivo experiments.

Cell line xenograft experiments were performed in 6 to 8-week-old NMRI mice (Charles River Laboratories) or NSG mice by injecting 10 million PC9 cells in 50% matrigel subcutaneously on each flank of the mouse. The different PDX models (EGFR[T790M/L858R] TP103[57], EGFR[ex20ins] LU0387[58], KRAS[G12C] TP60, KRAS[G12C] TP79)[59] were cut and engrafted subcutaneously into the flank of NSG mice or Nude mice for LU0387.

When tumors reached on average 200 to 300 mm³ (calculated as [length × width² ×3.14/6]), male and female mice were randomized and treated by oral gavage 5 days a week with 100 μl of vehicle or the appropriate treatment. For the LU0387 model, tipifarnib was administered either continuously or intermittently (1 week on, 1 week off). Tumors were measured twice a week. Mice were weighted twice a week

to monitor toxicity and humanely killed at indicated times and tumors were harvested. The maximal tumor size/burden authorized was 1500 mm³. The endpoints in this study were determined by the maximal tumor size, which was not exceeded, or the degradation of the general condition of the animal. PDX models are listed in Supplementary Table 5.

## Immunohistochemistry

Tissues were fixed in 10% formalin overnight at room temperature, stored in 70% ethanol at 4 °C and embedded in paraffin. Three-micrometer paraffin sections were used for hematoxylin and eosin staining followed by immuno-histochemistry. Ki67 staining was done with 1/500 D3B1 clone for one hour (Cell Signaling Technologies) after twenty minutes of high pH using PTlink (Agilent). The antibody was visualized using EnVision system in the Autostainer according to the manufacturer (Agilent). Hematoxylin was used as a counterstain. Two operators blindly evaluated quantification of Ki67 IHC scores. Images were retrieved using the NIS-Elements Viewer 5.21 software.

## Statistical analysis

When comparing two groups, statistical significance was calculated by two-sided unpaired t-test with Welch's correction or Wilcoxon tests. Statistical details and sample size can be found in the figures legend. Statistical analyses were performed using GraphPad Prism v9.0.0. Differential analysis and statistical significance of RNAseq data was performed using the DESeq2 method. Differential expression of scRNAseq data was performed as a pseudo-bulk analysis by randomly assigning an equal number of cells in 3 groups/replicates per condition, sum up the reads accordingly, and we finally performed a differential expression. For this purpose, we converted Seurat object to SingleCellExperiment and applied edgeR via scran package to process reads as a kind of classic RNAseq. In animal experiments, n represents the number of animals in treatment groups. Survival curves were estimated by the Kaplan-Meier method and the log-rank test using GraphPad Prism. Spearman's correlation analysis was performed using the cBioPortal software (www.cbioportal.org) on RSEM normalized Illumina HiSeq_RNAseqv2 data. For animal studies, animals were randomized before treatments, and all animals treated were included in the analyses. No statistical method was used to predetermine the sample size. No data were excluded from the analyses. The Investigators were not blinded to allocation during experiments and outcome assessment, except for the quantification of Ki67 IHC scores.

Pearson correlation coefficient (r) was determined by comparing the mean expression value per cluster of each genes, with a reference gene from each phenotype (e.g., *AGER* for the AT1 phenotype, *SERPINE1* for the mesenchymal phenotype, and *BPIFB1* for the mucous/serous phenotype). Genes that had a Pearson correlation coefficient higher than 0.9 were considered significantly representative of the variations observed for the reference gene and were used for subsequent GSEA analysis. Statistical analysis was performed by simple linear regression analysis, and all the genes with $r > 0.9$ had a p-value < 0.01.

## Reporting summary

Further information on research design is available in the Nature Portfolio Reporting Summary linked to this article.

## Data availability

The RNAseq and scRNAseq data generated in this study are publicly available at the NCBI Gene Expression Omnibus (GEO) database under accession codes GSE249721 and GSE248450, respectively. Other publicly available RNA-seq and scRNAseq transcriptomic data used in this study are available through the NCBI GEO database under accession codes GSE198672[23], GSE193259[46], GSE164326[47], GSE188406[48], and GSE64550[49]. TCGA expression data of human healthy lungs and lung

adenocarcinoma used in this study are available at http://firebrowse. org. The remaining data are available within the article, supplementary information or source data file. Source data are provided as a Source Data file. Source data are provided with this paper.

## Code availability

All software used are open source, Seurat and Deseq2 libraries are respectively available via CRAN and Bioconductor (https:// bioconductor.org/). Also in R environment, Dynverse packages are available via devtools as "dynverse/dyno". In addition, both scVelo[76] (based on python environment) and scSignatureExplorer[78] are available via at GitHub at https://github.com/theislab/scvelo and https:// github.com/FredPont/spatial, respectively.

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

## Acknowledgements

We thank all members of the SIGNATHER team at CRCT (Centre de Recherches en Cancérologie de Toulouse (CRCT), Inserm, CNRS, Université de Toulouse, Université Toulouse III Paul Sabatier, Toulouse, France) for stimulating and thoughtful discussions. We thank David Santamaría, John Hickman and Agnese Cristini for critically reading the manuscript. We thank all members of mice core facilities (CREFRE, UMS006 Inserm, Toulouse) in particular Cédric Baudelin, Amanda Corini, Emilie Sinhlivong, Tristan Chapon, Charlotte Delos, Thi-Lan Huong Huynh and Charlène Lopez for their support and technical assistance. We acknowledge the cytometry/cell-sorting and imaging facilities of the CRCT, in particular Manon Farcé and Laetitia Ligat for their assistance with flow cytometry and microscopy, respectively. We thank the Genomics and Transcritomics platform of the CRCT especially Carine Valle and Emeline Sarot for their support and technical assistance, Toulouse GENOTOUL for sequence alignments, as well as the Bioinformatics platform of the CRCT, especially Marie Tosolini for the bioinformatics analysis. We thank Géraldine Touriol for the administrative and financial management. This work was supported in part by institutional grants from Inserm, CNRS, Fondation pour la Recherche Médicale (FRM, équipe labellisée [DEQ20170839117]), Programme de Recherche Translationnelle en Cancérologie (INCa-DGOS, [PRT-K18-048], J.M.), Fondation Toulouse Cancer Santé (FTCS, O.C. and J.M.), Labex TOUCAN (O.C.), Ligue Nationale Contre le Cancer (LNCC, GF), Fondation ARC (PJA2, O.C.; PhD, S.F.), ALK+ROS1 France patient association (O.C.), Kura Oncology sponsored research (O.C.).

## Author contributions

Study design, data interpretation and preparation of the manuscript: S.F., C.D., R.G., A.P., J.M., G.F. and O.C.; Execution of experiments: S.F.,

C.D., R.G., A.D., M.B., C.T., N.B., R.A., V.D., C.M., A.F., S.P., E.T.C., A.C., O.C.; Computational and statistical analysis: J.C., J.P.V., J.P.C., O.C.; Writing, review, and/or revision of the manuscript: S.F., C.D., R.G., J.P.C., N.B., J.P.V., A.M., A.P., L.K., F.B., J.M., G.F., O.C.; Material support: I.L.M., I.F., L.P.A.

## Competing interests

O.C. reports research funding from Kura Oncology through a sponsored research contract. L.K. reports personal fees from Kura Oncology during the conduct of the study and personal fees from Kura Oncology outside the submitted work. FB is an employee and stockholder of Kura Oncology. JM reports personal fees/advisory board membership from Roche and Bristol Myers Squibb, and AstraZeneca, advisory board membership and research funding (institution), personal fees/advisory board membership from Pfizer, Novartis, Amgen, Takeda, Daiichi Sankyo, the healthcare business of Merck KGaA, Darmstadt, Germany, grants/funding (institution) from Roche/Genentech, Bristol Myers Squibb, Pierre Fabre and AstraZeneca outside the submitted work. L.P.A has leadership interest (board member) in ALTUM Sequencing and Genomica; has received honoraria for participation at meetings from Amgen Sanofi, AstraZeneca Spain, Bayer, Blueprint Medicines, Bristol Myers Squibb/Celgene, Daiichi Sankyo, Ipsen, Lilly, Merck Serono, Mirati Therapeutics, Novartis, Pfizer, PharmaMar, Roche/Genentech, Servier, and Takeda; speakers' bureau from AstraZeneca, Bristol Myers Squibb, Merck Serono, MSD Oncology, Pfizer, and Roche/Genentech; research funding (via his institution) from AstraZeneca, Bristol Myers Squibb, Kura Oncology, MSD, Pfizer and PharmaMar; and travel and accommodation expenses from AstraZeneca, Bristol Myers Squibb/Celgene, MSD, Pfizer Roche/Genentech, and Takeda. L.P-A. also declares other relationships with Amgen, Ipsen, Merck, Novartis, Pfizer, Roche, Sanofi, and Servier (as sponsors of clinical trials), outside the submitted work. The remaining authors declare no conflict of interest.

## Additional information

[1]Centre de Recherches en Cancérologie de Toulouse (CRCT), Inserm, CNRS, Université de Toulouse, Université Toulouse III Paul Sabatier, Toulouse, France. [2]Institut de Recherche en Cancérologie de Montpellier (IRCM), Inserm, Université de Montpellier, Institut Régional du Cancer de Montpellier (ICM), Montpellier, France. [3]Unidad de Investigación Clínica de Cáncer de Pulmón, Instituto de Investigación Hospital 12 de Octubre-CNIO, Madrid, Spain. [4]Kura Oncology, Inc, Calif, USA. [5]Centre Hospitalier Universitaire (CHU) de Toulouse, service de pneumologie, Toulouse, France. [6]Oncopole Claudius Regaud, Institut Universitaire du Cancer de Toulouse-Oncopole, Laboratoire de Biologie Médicale Oncologique, Toulouse, France. [7]These authors contributed equally: Sarah Figarol, Célia Delahaye, Gilles Favre, Olivier Calvayrac. ✉e-mail: favre.gilles@iuct-oncopole.fr; olivier.calvayrac@inserm.fr

