## [Peer Review File · Nature Communications]

Reviewers' Comments:

Reviewer #1:

Remarks to the Author:

The authors have submitted a revised manuscript. The manuscript is improved. There are several limitations to the study that should be discussed in a limitations paragraph in the discussion. These include the specificity of the FTI effect for the DTP state, the target(s)/mechanism of the FTI effect, and the lack of deep mechanistic insight into the relationship between the different forms of lineage plasticity and the FTI effects.

Overall, the authors have addressed the comments of each reviewer, including reviewer 3. However, the authors should temper the main claims in the final text and include clear written acknowledgement of the limitations of the findings highlighted by reviewer 3 and each reviewer.

Reviewer #2:

Remarks to the Author:

The authors have addressed 4 out of 5 points I raised and provide some additional discussion of the fifth point, but not experimental clarification. Overall, I consider this to be a satisfactory response and firmly recommend publication of the study. It is a large and detailed body of work with some impressive pre-clinical data.

Reviewer #1 (Remarks to the Author):

The authors have submitted a revised manuscript. The manuscript is improved. There are several limitations to the study that should be discussed in a limitations paragraph in the discussion. These include the specificity of the FTI effect for the DTP state, the target(s)/mechanism of the FTI effect, and the lack of deep mechanistic insight into the relationship between the different forms of lineage plasticity and the FTI effects.

Overall, the authors have addressed the comments of each reviewer, including reviewer 3. However, the authors should temper the main claims in the final text and include clear written acknowledgement of the limitations of the findings highlighted by reviewer 3 and each reviewer.

We thank the reviewer for the positive and constructive feedback, which helped us to substantially improve our manuscript. We acknowledge that our work have some limitations, which have been clearly highlighted and discussed in a dedicated paragraph in the Discussion section of this revised version:

L438-448: "There are some limitations in this study. First, the relationship between lineage plasticity during the adaptive response to TT and the vulnerability to farnesyltransferase inhibition is still unknown. This mechanistic gap between the hallmarks of drug tolerance and the FTIs effects partially stems from the lack of formal identification of the farnesylated proteins involved in the phenotypic reprogramming and responsible for the relapse. Second, the mechanism by which FTI treatment induces the integrated stress response has not been elucidated in this work, and should be further investigated. Finally, the precise therapeutic window for the co-treatment of targeted therapies and FTIs, and the specificity of the FTI effect for the drug-tolerant state, remain to be clarified. This is probably due to a lack of a clear definition of the drug-tolerant state, particularly defining when drug tolerance begins and when resistance takes over. Additional experiments are required to understand why some early relapsing or fully resistant proliferative cells retain sensitivity to FTI, and why some others do not."

Moreover, we also toned down the main claims in the manuscript regarding the phenotypic transition from alveolar-type 1 population to the mesenchymal phenotype in the results section:

L135: "The drug-tolerant state displayed two distinct although tightly linked populations"

L138-142: "The sequential emergence of the two phenotypes, first alveolar and then mesenchymal, combined with pseudotime analysis, suggest that mesenchymal cells may have evolved from the alveolar-like population. However, we cannot exclude that tumor cells could have undergone distinct paths of drug adaptation toward both phenotypes during the adaptive response."

L364-368: "Although our study cannot exclude that AT1-like and mesenchymal cells could have undergone distinct paths of drug adaptation at some point during the adaptive response, the timing of emergence of both populations and their tight phenotypic similarities revealed by scRNAseq suggest that the alveolar-like population may act as a reservoir from which mesenchymal cells could emerge."

Finally, we globally refrained from using exaggerated language throughout our manuscript.

Reviewer #2 (Remarks to the Author):

The authors have addressed 4 out of 5 points I raised and provide some additional discussion of the fifth point, but not experimental clarification. Overall, I consider this to be a satisfactory response and firmly recommend publication of the study. It is a large and detailed body of work with some impressive pre-clinical data.

We thank the reviewer for the positive and constructive feedback, which helped us to substantially improve our manuscript. An additional paragraph has been added to the Discussion section to highlight and discuss the limitations of our study.